# Comparative Transcriptome Analyses Provide Potential Insights into the Molecular Mechanisms of Astaxanthin in the Protection against Alcoholic Liver Disease in Mice

**DOI:** 10.3390/md17030181

**Published:** 2019-03-19

**Authors:** Huilin Liu, Huimin Liu, Lingyu Zhu, Ziqi Zhang, Xin Zheng, Jingsheng Liu, Xueqi Fu

**Affiliations:** 1College of Life Science, Jilin University, Changchun 130012, China; ireneliuhl@163.com; 2College of Food Science and Engineering, Jilin Agricultural University, Changchun 130118, China; liuhuimin@jlau.edu.cn; 3National Engineering Laboratory for Wheat and Corn Deep Processing, Changchun 130118, China; 4College of Animal Science and Technology, Jilin Agricultural University, Changchun 130118, China; 13610728530@163.com (L.Z.); zzqknight@163.com (Z.Z.); zhengxin@jlau.edu.cn (X.Z.)

**Keywords:** astaxanthin, comparative transcriptome analyses, alcoholic liver disease, bioinformatic analysis

## Abstract

Alcoholic liver disease (ALD) is a major cause of chronic liver disease worldwide. It is a complex process, including a broad spectrum of hepatic lesions from fibrosis to cirrhosis. Our previous study suggested that astaxanthin (AST) could alleviate the hepatic inflammation and lipid dysmetabolism induced by ethanol administration. In this study, a total of 48 male C57BL/6J mice were divided into 4 groups: a Con group (fed with a Lieber–DeCarli liquid diet), an AST group (fed with a Lieber–DeCarli liquid diet and AST), an Et group (fed with an ethanol-containing Lieber–DeCarli liquid diet), and a EtAST group (fed with an ethanol-containing Lieber–DeCarli liquid diet and AST). Then, comparative hepatic transcriptome analysis among the groups was performed by Illumina RNA sequencing. Gene enrichment analysis was conducted to identify pathways affected by the differentially expressed genes. Changes of the top genes were verified by quantitative real-time PCR (qRT-PCR) and Western blot. A total of 514.95 ± 6.89, 546.02 ± 15.93, 576.06 ± 21.01, and 690.85 ± 54.14 million clean reads were obtained for the Con, AST, Et, and EtAST groups, respectively. Compared with the Et group, 1892 differentially expressed genes (DEGs) (including 351 upregulated and 1541 downregulated genes) were identified in the AST group, 1724 differentially expressed genes (including 233 upregulated and 1491 downregulated genes) were identified in the Con group, and 1718 DEGs (including 1380 upregulated and 338 downregulated genes) were identified in the EtAST group. The enrichment analyses revealed that the chemokine signaling, the antigen processing and presentation, the nucleotide-binding and oligomerization domain (NOD)-like receptor signaling, and the Toll-like receptor signaling pathways enriched the most differentially expressed genes. The findings of this study provide insights for the development of nutrition-related therapeutics for ALD.

## 1. Introduction

A recent report from the World Health Organization indicates that three million deaths every year result from the harmful use of alcohol (representing 5.3% of all global deaths) [1,2]. Alcohol abuse is one of the leading causes of more than 200 disease and injury conditions worldwide [3]. As the liver is the major organism for the metabolism of alcohol, long-term use and over-consumption of alcohol leads to alcoholic liver disease (ALD). ALD is a complex process including a broad clinical-histologic spectrum of hepatic lesions, ranging from simple fatty liver to liver injury, from steatosis to cirrhosis, and even hepatocellular carcinoma [3,4]. The pathogenesis of ALD has been well characterized, but no specific drugs and therapy were available to reverse this progress in humans. Emerging evidence showed that inflammation, oxidative stress, cell injury, and regeneration are predominant drivers for ALD [5,6]. Recent studies also showed that some intracellular signaling pathways and transcriptional factors are involved in the mechanisms of pathogenesis of ALD [7].

Astaxanthin (AST) is a significant xanthophyll carotenoid, mainly derived from marine organisms and algae; it cannot be synthesized in humans. AST is a lipid-soluble compound which cannot be converted to vitamin A in the human body [8]. Growing research indicates that AST provides many benefits to humans, such as antioxidant effects, anti-apoptosis effects, anti-inflammation effects, neuroprotective effects, cardiovascular disease prevention, and immune-modulation effects [9,10]. Notably, it is a potential protector against liver damage, such as liver fibrosis and non-alcoholic fatty liver disease. However, previously published studies are limited to the protective effect of AST on ALD [11,12]. In vivo studies have suggested that decreased AST serum levels of aspartate transaminase and alanine transaminase in the livers of the AST administrated group could alleviate the hepatic inflammation and lipid dysmetabolism induced by ethanol administration [12,13]. However, the molecular mechanisms of AST in the protective effect of ALD are still unclear. Hence, in the present study, a genome-wide comparison of the transcriptome with or without AST in ALD mice was done using RNA-sequencing (RNA-Seq) analysis. Then, differential protein expression was confirmed using immunoblot. This provides insight on how astaxanthin affects ALD and gives evidence on the molecular mechanism of AST in ALD protection.

## 2. Results

### 2.1. Overview of RNA-Sequencing Analysis 

After removing the low-quality reads and quality control, a total of 514.95 ± 6.89, 546.02 ± 15.93, 576.06 ± 21.01, and 690.85 ± 54.14 million clean reads were obtained for the Con, AST, Et, and EtAST groups, respectively (see Table 1). The clean GC content of each group ranged from 47.78 to 49.2%, and the value of Q30 ranged from 92.11 to 93.56% (see Appendix A). To evaluate the quality of the RNA-Seq data, the total clean reads were mapped to the reference genome. A high proportion of the clean reads were mapped to the mouse reference genome using Tophat2 (http://tophat.cbcb.umd.edu/); that is, 92.02% from Con, 91.85% from AST, 91.59% from Et, and 91.5% from EtAST (see Table 1). Through nucleotide basic local alignment search tool (BLAST) analysis, more than 97% of the reads of each group were mapped to known genes, and more than 95% of the reads were mapped to exons. Furthermore, principal component analysis revealed high correlations among biological replicates (Appendix A). Together, all the results indicated that the RNA-Seq data was reliable.

### 2.2. Gene Annotation and Functional Analysis

The genes were aligned with public databases, such as the Gene Ontology (GO) database, the Kyoto Encyclopedia of Genes and Genomes (KEGG), and eggNOG (i.e., the Evolutionary Genealogy of Genes: Non-supervised Orthologous Groups). As shown in Table 2, most of the genes were annotated using the GO database (97.06%), followed by eggNOG (74.58%) and KEGG (61.52%).

GO is an international standardized gene functional classification system. In total, there were 21,430 genes mapped in the GO database (Appendix A). The biological process group possessed more terms than the cellular component and molecular function groups. The highly enriched GO terms were in the cellular process (GO: 0009987), biological regulation (GO: 0065007), metabolic process (GO: 0008152), response stimulus (GO: 0050896), multicellular organismal process (GO: 0032501), and signaling (GO: 0023052) groups.

Furthermore, the genes were annotated and classified using the KEGG database. As shown in Appendix A, genes assigned to human diseases (2705) occupied the maximum proportion, followed by those assigned to signal transduction (1784) and cellular processes (1728).

### 2.3. Identification of Differentially Expressed Genes (DEGs)

Gene expression levels of Con, AST, Et, and EtAST were quantified and compared (Figure 1). The genes with a reads per kilobases per million (RPKM) ratio greater than twofold were defined as DEGs. As shown in Figure 1a, a total of 15,779, 15,740, 16,136, and 15,877 DEGs were identified in the Con, AST, Et, and EtAST groups, respectively. Among these DEGs, there were 168, 163, 351, and 201 DEGs uniquely expressed in Con, AST, Et, and EtAST, respectively. Moreover, 14,917 DEGs were commonly expressed in all the groups.

Significant DEGs, including upregulated or downregulated genes, were identified by DEGseq (Figure 1B). Compared with the Et group, 1892 DEGs, including 351 upregulated and 1541 downregulated genes, were identified in the AST group; 1724 DEGs, including 233 upregulated and 1491 downregulated genes, were identified in the Con group; and 1718 DEGs, including 1380 upregulated and 338 downregulated genes, were found in the EtAST group.

To confirm the gene expression level acquired by RNA-Seq, 16 genes were determined by quantitative real-time PCR (qRT-PCR). The results showed that the expression levels of the 16 genes were consistent with the RNA-Seq data (Figure 2), which indicated that the RNA-Seq data in the present study were reliable.

### 2.4. KEGG Enrichment Analyses of DEGs

To uncover the potential mechanisms of AST involved in the protective effect against ALD, we performed KEGG enrichment analyses using a path-finder software. In the previous study, we found that AST could influence the immune system to ameliorate liver injury [12]. Thus, to identify the specific biological pathways involved in the immune system, we conducted KEGG enrichment analyses on pairs of comparison groups (Con versus Et and Et versus EtAST). The top 10 ranked KEGG pathways for each comparison group are summarized in Table 3 and Table 4. 

As shown in Table 3, compared with Con mice, most of the DEGs related to the immune system were up-regulated in Et mice. It indicated that alcohol consumption disrupts the immune system in complex ways. However, AST administration could ameliorate these disruptions (Table 4). Between Con and Et mice, the chemokine signaling pathway (ko04062), the antigen processing and presentation pathway (ko04612), and the NOD-like receptor signaling pathway (ko04621) were the most significantly enriched pathways (Table 3). To Et and EtAST mice, the top 3 pathways were the natural killer cell mediated cytotoxicity pathway (ko04650), the NOD-like receptor signaling pathway (ko04621), and the chemokine signaling pathway (ko04062) (see Table 4). Together, the chemokine signaling pathway, the NOD-like receptor signaling pathway, and the Toll-like receptor signaling pathway were chosen for further validation.

### 2.5. qRT-PCR Validation of Differentially Expressed Genes

To verify the results of the transcriptome sequencing and further analyze the key gene expressions involved in astaxanthin regulating alcoholic liver disease, 13 representative genes were selected from the chemokine signaling pathway, the NOD-like receptor signaling pathway, and the Toll-like receptor signaling pathway, and were quantified by qRT-PCR. 

As shown in Figure 3A, ethanol significantly upregulated the expression of certain genes, including Interleukin-1 alpha (IL-1α), Interleukin-1 beta (IL-1β), PYD domains-containing protein (NLRP3), Caspase1, and Interleukin-18 (IL-18) in the Et group, compared with Con group. However, AST supplement in the EtAST group reversed this effect, and showed no significant difference compared with the Con group

As shown in Figure 3B, the detection of the representative genes in the Toll-like receptor signal pathway—including Toll-like receptors 2, 3, 4, and 6 (TLR2, 3, 4, 6) and myeloid differential protein-88 (MyD88)—were significantly upregulated in the Et group, compared to the Con and AST groups, whereas an AST supplement in the EtAST group reversed this effect, which showed no difference compared to the Con group. 

As shown in Figure 3C, compared with the Con group, ethanol significantly upregulated the representative genes from the chemokine signaling pathway, including the monocyte chemoattractant protein-1 (MCP-1) and macrophage inflammatory protein 2 (MIP-2). AST significantly downregulated the two genes, but showed no significant difference compared with the Con group. The qRT-PCR results showed a similar downregulated trend with the gene expression found through RNA-Seq, and the coincidence rate was more than 82%; therefore, the qRT-PCR expression validates the findings of RNA-Seq.

Overall, these results suggest that AST reversed the inflammation caused by ethanol through the regulated chemokine signaling pathway, the NOD-like receptor signaling pathway, and the Toll-like receptor signaling pathway.

### 2.6. Western Blot Validation of Differentially Expressed Genes

To further investigate the mechanism underlying the hepatoprotective effects of AST on alcohol-induced liver inflammation, we examined the protein expression levels of the Toll-like receptor and NOD-like receptor. Compared with the Et group, the protein levels of MYD88, TLR4, NLRP3, and IL-1β were significantly decreased in the Con and EtAST groups. However, there was no significant difference in the levels of MYD88, TLR4, and IL-1β in the AST group (Figure 4). It has been reported that the Toll-like receptor and the NOD-like receptor were relevant to the NF-κB and MAPK families. Next, the representative proteins—including JNK, p38, ERK 1/2, and p65—involved in these two families were detected. The phosphorylation levels of JNK, p38, ERK 1/2, and p65 were significantly increased in the Et group when compared with the Con group, and these proteins decreased in level after the AST supplement was administered, compared with the Et group (Figure 4). Taken together, these results suggest that AST has protective effects on alcoholic liver injury and causes an associated depression in the expression of p65, JNK, p38, and ERK1/2.

## 3. Discussion

AST is similar to β-carotene in molecular structure and possesses a strong antioxidative effect [10]. Recently, researchers have shown an increased interest in AST due to the demand in the promotion of human health [9]. Previous research has established that AST can relieve ischemia-related brain injury by suppressing oxidative stress [14], exerting neuroprotective effects by weakening neuroinflammation [15], and modulating the endogenous antioxidant defense system [16]. Moreover, AST is also a potential protector against liver damage [11]. It can inhibit liver fibrosis and lipid peroxidation [17]; inhibit liver tumorigenesis and inflammation [18]; attenuate hepatic ischemia reperfusion-induced apoptosis and autophagy [19,20]; and prevent ethanol-induced hepatic injury through the in vivo inhibition of oxidant and inflammatory responses [21]. Additionally, AST possesses anti-fibrogenic effects, through the TGFβ1–Smad3 signaling pathway in hepatic stellate cells [22]. However, most of the in vitro studies were performed on hepatocellular carcinoma cells and hepatic stellate cells. In term of the in vivo studies, nonalcoholic steatohepatitis mice, diabetic mice, and obese mice were the major mouse models. Thus far, very little attention has been paid to the molecular mechanisms of ALD nutrition prevention and protection. Our previous study indicated that AST administration significantly relieves inflammation, decreases lipid accumulation, and improves serum marker levels relating to ethanol-induced liver injury [12]. The present study was designed to determine possible molecular mechanisms involved in the protective effect of AST in ALD. We found that AST may prevent the progress of ALD through the chemokine signaling pathway, the NOD-like receptor signaling pathway, and the TLR signaling pathway (Figure 3 and Figure 4).

Currently, alcohol abuse is one of the major drivers of chronic liver disease in Western countries [2]. Growing evidence suggests that TLRs play a vital role in the pathogenesis and progression of liver diseases, such as alcoholic liver disease (ALD), non-alcoholic fatty liver disease (NAFLD), and autoimmune liver disease [23]. TLRs belong to a family of pattern recognition receptors that recognize pathogen-associated molecular patterns (PAMPs) and damage-associated patterns (DAMPs). They play an important role in the initiation of the immune system and inflammation process [24]. Once activated, TLRs are expressed by liver-resident cells and trigger the production of cytokines and chemokines [23]. The mRNA expression levels of TLRs is deficit in healthy liver cells, and it seems that the TLR-signaling pathway is not activated [25]. As shown in Figure 3A, compared with Et mice, the mRNA expression levels of TLR2 and TLR4 were very low in normal mice. The protein level of TLR4 showed the same tendency (see Figure 4). Myeloid differentiation primary response 88 (MYD88) is the canonical adaptor for inflammatory signaling pathways downstream from members of the TLRs [26]. It is suggested that the TLR4 downstream signaling is regulated by the MyD88-independent pathway in ALD [26]. Both mRNA and the protein level of MyD88 were examined in the present study (see Figure 3 and Figure 4). These results are in line with the previous study and provide further support for the hypothesis that TLRs participate in the progress of ALD, and could be a therapeutic target for the treatment of ALD.

Activation of the immune signaling pathways plays a critical role in the pathogenesis of ALD [27]. NOD-like receptors (NLRs) are intracellular innate immune sensors that could recognize pathogen-PAMPs and -DAMPs [28]. Recent studies suggest that NLRs are not only expressed and activated in innate immune cells, but also in parenchymal cells in the liver. NLRs can work with TLRs and regulate the inflammatory response [29]. The NLRs consist of two major subfamilies, NODs and NLRPs, containing 23 members in humans and 34 in mice [28]. NLRs are key mediators of the inflammasome, which are the major drivers of inflammation. There are several inflammasomes, such as NLRP1, NLRP3, and NLRC4. Thus far, the most characterized and investigated member is NLRP3 [30]. Recently, it was demonstrated that the NLRP3 inflammasome was involved in the development of chronic liver diseases, such as alcoholic steatohepatitis and NAFLD. Once activated, it upregulates the expression of caspase-1, then promotes the secretion of IL-1β and IL-18, which play key roles in the induction and progression of liver inflammation [31,32]. The results of this study indicate that AST administration can significantly downregulate the expression of NLRP3, but not NLRP1. Thus, the mRNA levels of downstream factors, such as caspase-1 and IL-18, were decreased. Compared with Et mice, AST treatment notably downregulated the hepatic protein expression of NLRP3 and IL-1β in EtAST mice (Figure 3 and Figure 4). Taken together, these findings suggest that AST may protect against alcohol-induced liver injury through the NOD pathway.

As mentioned above, innate immune cells play a key role in the pathophysiology of ALD. Resident macrophages in the liver are activated by PAMPs and DAMPs. The expression of TLRs and NLRs are then stimulated. This, finally, triggers the production of cytokines and chemokines [33]. It was revealed that chemokines were important determinants in the pathogenesis of liver disease. They regulated the migration and activities of the resident cells in the liver [33,34]. Chemokines were initially discovered for their role in regulating cell trafficking. To date, more than 50 chemokine ligands and 19 receptors have been identified, several of which have been described as relevant in the progress of liver disease [35]. Monocyte chemoattractant protein-1 (MCP1) and macrophage inflammatory protein-2 (MIP2) were the most-studied chemokines involved in ALD. MCP1, also known as CCL2, regulates macrophage activation, proinflammatory responses, and hepatic steatosis in the liver [36]. Previous studies have indicated that the plasma and hepatic levels of MCP1 were elevated in ALD patients. Moreover, it was shown that the deficiency of MCP1 protects against alcoholic liver injury by inhibiting the production of proinflammatory cytokines in mice. All these findings suggested that MCP1 might be a potential therapeutic target in ALD [36,37,38]. In Figure 3, it is evident that there is a significant difference in the hepatic expression levels of MCP1 between Et and EtAST mice. MIP2, also named CXCL2, is mainly activated by Kupffer cells in liver injury. It accelerates liver inflammation by releasing various inflammatory mediators [39]. In vivo studies have indicated that plasma and hepatic MIP2 concentrations were increased in ALD mice [40]. In our study, the hepatic mRNA expression of EtAST mice was dramatically lower compared to Et mice (see Figure 3). According to this data, we can infer that AST could improve hepatic inflammation by inhibiting the expression of MCP1 and MIP2 in ALD mice or patients.

## 4. Materials and Methods 

### 4.1. Animal Experimentation

Male C57BL/6J mice (20–24 g, six-weeks-old) were purchased from the Beijing Vital River Laboratory Animal Technology Co., Ltd (Beijing, China). Mice were housed individually in cages for a 12 h light/dark cycle at 23 ± 2 °C with optimum access to chow and water ad libitum. After one-week acclimation, a total of 48 mice were randomly divided into four groups: (1) the Con group (*n* = 12), given a Lieber–DeCarli liquid diet (Appendix A) for 12 weeks; (2) the AST group (*n* = 12), given a Lieber–DeCarli liquid diet for the first two weeks, then a Lieber–DeCarli liquid diet with astaxanthin (AST, 50 mg/kg body weight) for another 10 weeks; (3) the Et group (*n* = 12), given a Lieber–DeCarli liquid diet for the first two weeks, then an ethanol-containing Lieber–DeCarli liquid diet (i.e., 5% ethanol *v/v* accounted for 36% the total caloric intake) for another 10 weeks; and (4) the EtAST group (*n* = 12), given a Lieber–DeCarli liquid diet for the first two weeks, then an ethanol-containing Lieber–DeCarli liquid diet plus astaxanthin (50 mg/kg body weight) for another 10 weeks. The AST was purchased from Sigma–Aldrich (St Louis, MO, USA. SML0982) and dissolved in corn oil for further use. The AST dose was done with reference to the previous research [41], while the ethanol uptake amount was increased for two weeks, and the final concentration was 5% (*v/v*). After the mice were sacrificed, the liver tissues were collected and frozen in liquid nitrogen overnight for RNA extraction. All experiment protocols were approved by the Institutional Animal Care and Use Committee at the Jilin Institute of Traditional Chinese Medicine (Approval Number SYXK (JI) 2015-0009).

### 4.2. RNA Sequencing

Total fresh liver tissues were suspended in TRIzol reagent (Invitrogen Life Technologies, Carlsbad, CA, USA) according to the manufacturer’s protocol. The concentration and purity of RNA was determined using a NanoDrop 2000 microspectrophotometer (Thermo Fisher Scientific, Waltham, MA, USA).

The sequencing libraries were generated using the TruSeq RNA Sample Preparation Kit (Illumina, San Diego, CA, USA), which consists of an mRNA purification process that uses poly-T beads, mRNA fragmentation, reverse transcription, end repair, the addition of a single ‘A’ base, ligation of the adapters, and purification and enrichment with PCR. The library fragments were purified using the AMPure XP system (Beckman Coulter, Beverly, CA, USA) and quantified using the Agilent High Sensitivity DNA assay on a Bioanalyzer 2100 system (Agilent, Santa Clara, CA, USA). Sequencing was carried out using an Illumina Hiseq Xten platform (150 bp paired-end reads).

### 4.3. Differential Expression Gene Analysis 

The gene expression level was quantified using RPKM and analyzed using the HTSeq software (Version 0.6.1p2, http://www-huber.embl.de/users/anders/HTSeq), using union as the counting model. A cut-off value of RPKM > 1 was used to define the gene expression. DEG analysis was performed using DESeq (Version 1.18.0), and the fold change and Fisher-test were used to choose differentially expressed genes [42]. The false discovery rate (FDR) criterion was introduced to adjust the *p*-values. In this study, the differential genes with the *p*-value < 0.05 and the false discovery rate (FDR) < 0.02 were considered to be statistically significant. 

### 4.4. The Enrichment Analyses of Differentially Expressed Genes

Differentially expressed gene enrichment analyses were performed using KEGG databases. KEGG is a database resource dealing with genomes, biological pathways, diseases, drugs, and chemical substances. Here, KEGG enrichment analyses was conducted using the online Path-Finder software (http://www.genome.jp).

### 4.5. Quantitative Real-Time PCR 

qRT-PCR was performed to validate the DEGs obtained from the RNA-Seq results. Specific primers of 16 candidate DEGs were designed using the Primer 5 software (Version 5.0, Premier biosoft, Palo Alto, CA, USA)., and were synthesized by Sheng Gong (Shanghai, China) (Appendix A). Total RNA was extracted as previously described and reverse-transcribed using the PrimeScript™ RT Reagent Kit with a gDNA eraser (Takara, Tokyo, Japan). SYBR Green Mix (Takara, Tokyo, Japan) and a CFX96 Real Time PCR System (Bio-Red Laboratories, Hercules, CA, USA) were used to perform the qRT-PCR; following the manufacturer’s protocol, the melting curves were as follows: 95 °C for 60 s, followed by 55 °C for 30 s, and then 95 °C for 30 s. For an endogenous reference gene, 18 s rRNA was used. The mRNA expression levels were calculated using the 2^−ΔΔCt^ method. The QPCR assays were performed as compliant with MIQE.

### 4.6. Western Blotting

Western blotting was performed to evaluate the protein expression levels of DEGs. The liver tissues were homogenized using a liquid nitrogen pre-cooled high-speed tissue homogenizer (Gering Scientific Instruments Ltd, Beijing, China) and lysed using 10 μM phenylmethanesulfonyl fluoride (PMSF, Beyotime Institute of Biotechnology, Jiangsu, China) and a 1% protease inhibitor cocktail (104 mM AEBSF, 80 μM Aprotinin, 4 mM Bestatin, 1.4 mM E-64, 2 mM Leupeptin, and 1.5 mM Pepstatin A; Sigma). The protein contents were analyzed with the BCA Protein Assay Kit (Vazyme Biotech Co., Ltd, Nanjing, China). Standard Western blotting was performed, and the blots were visualized via a chemiluminescent system using ImageQuant LAS 500 imaging instruments (GE Healthcare Life Sciences, Shanghai, China) and quantified using the Image J analyzer software. Antibodies against ERK 1/2, p-ERK1/2, p-NF-kB p65, MAPK p38, p-MAPK p38, JNK, and p-JNK were purchased from Cell Signaling Technology (Danver, MA, USA). Antibodies against TLR4, NLRP3, IL-1b, MYD88, and NF-kB p65 were obtained from Abcam (Cambridge, MA, USA).

### 4.7. Statistical Analysis 

Statistical significance in the DEG analyses was performed using the R statistical package. Values of *p* < 0.05 were considered statistically significant.

The qRT-PCR and Western blot results were presented as means ± SEM and calculated using GraphPad Prism version 7.01 (GraphPad Software, Inc., La Jolla, CA, USA). Statistical significance was determined using the Tukey’s multiple-comparison test. Values of *p* < 0.05 were considered statistically significant.

## 5. Conclusions

In this study, comparative hepatic transcriptome analyses were performed to elucidate the possible mechanisms of AST in the protection against ALD. Through RNA-Seq, a total of 22,078 genes were identified. Then, KEGG enrichment analyses were conducted to reveal potential signaling pathways related to the immune system. We found that most DEGs were enriched in the chemokine signaling pathway, the NOD-like receptor signaling pathway, and the Toll-like receptor signaling pathway. Furthermore, 13 genes associated with these three pathways were selected to identify the RNA-sequencing results. This is the first report, to our knowledge, describing comparative transcriptome analyses of AST in the protection against ALD. We found that AST may prevent the progress of ALD through the chemokine signaling pathway, the NOD-like receptor signaling pathway, and the TLR signaling pathway. Our findings provide important insights into the possible molecular mechanisms of AST in the nutritional intervention of ALD.

## Figures and Tables

**Figure 1 marinedrugs-17-00181-f001:**
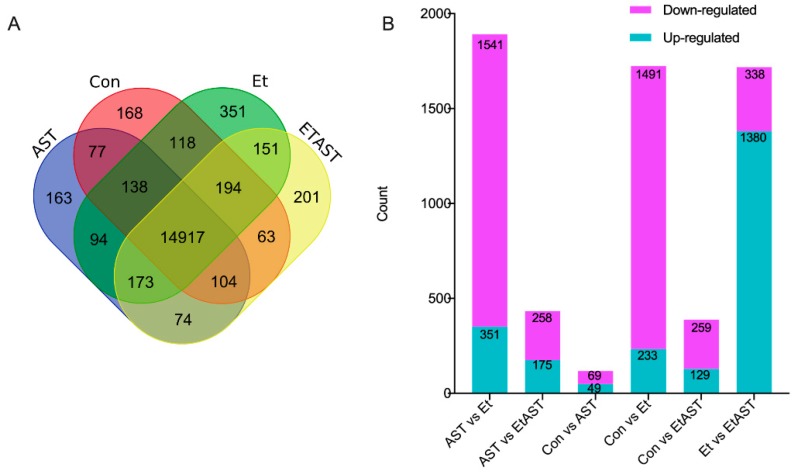
Statistical analysis of the gene expression detected by RNA-sequencing (RNA-Seq). (**A**) Venn diagram of gene counts expressed in the Con, AST, Et, and EtAST groups. (**B**) Number of total differentially expressed genes (DEGs) and down- or upregulated DEGs, respectively.

**Figure 2 marinedrugs-17-00181-f002:**
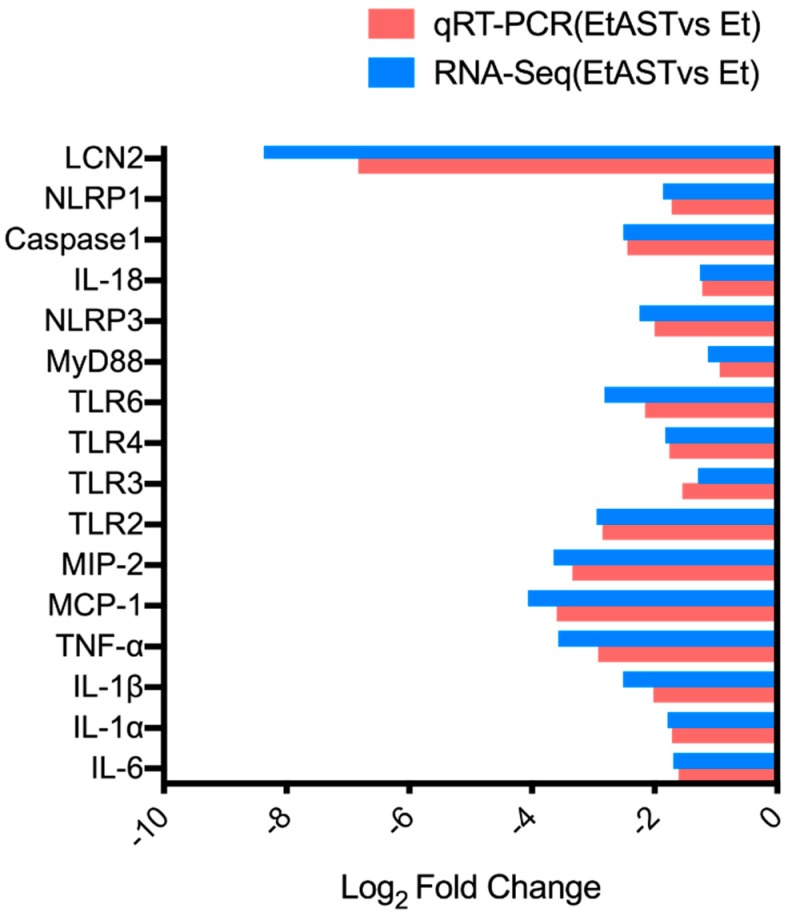
Quantitative real-time PCR (qRT-PCR) verification of RNA-sequencing results. The x-axis represents genes; the y-axis represents the logarithm of fold change; and the red column and blue column represents the qRT-PCR results and RNA-sequencing results, respectively.

**Figure 3 marinedrugs-17-00181-f003:**
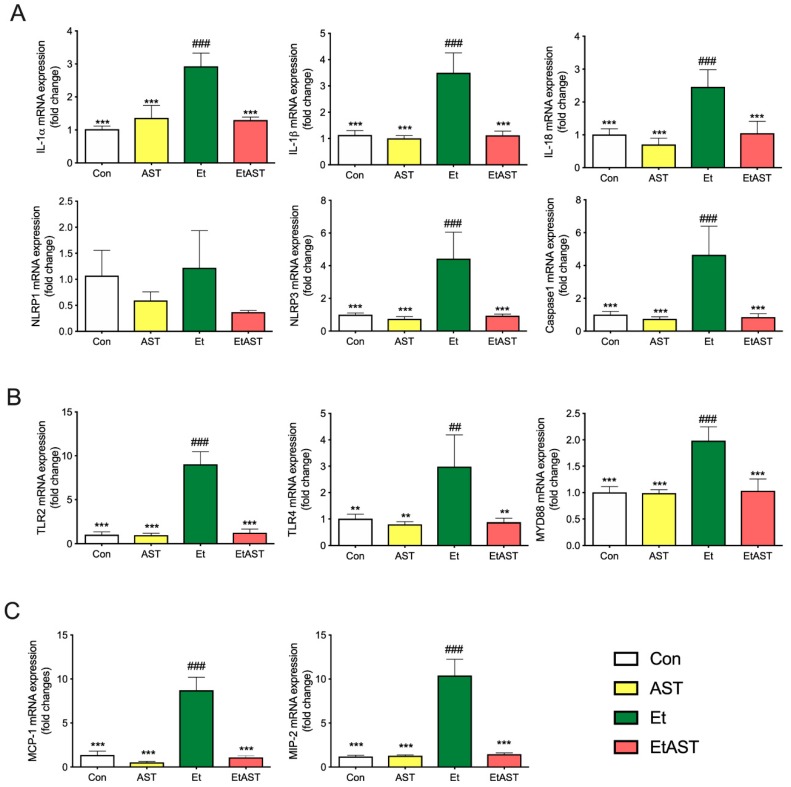
Hepatic mRNA expression levels of associated genes in (**A**) the NOD-like pathway, (**B**) Toll-like pathway, and (**C**) chemokines pathway in the Con, AST, Et, and EtAST groups. Relative mRNA expression levels were determined by real-time RT-PCR and normalized to 18s rRNA as an internal control. Data was represented as means ± SEM (*n* = 6; ** *p* < 0.01 versus Et; *** *p* < 0.001 versus Et; ## *p* < 0.01 versus Con; and ### *p* < 0.001 versus Con).

**Figure 4 marinedrugs-17-00181-f004:**
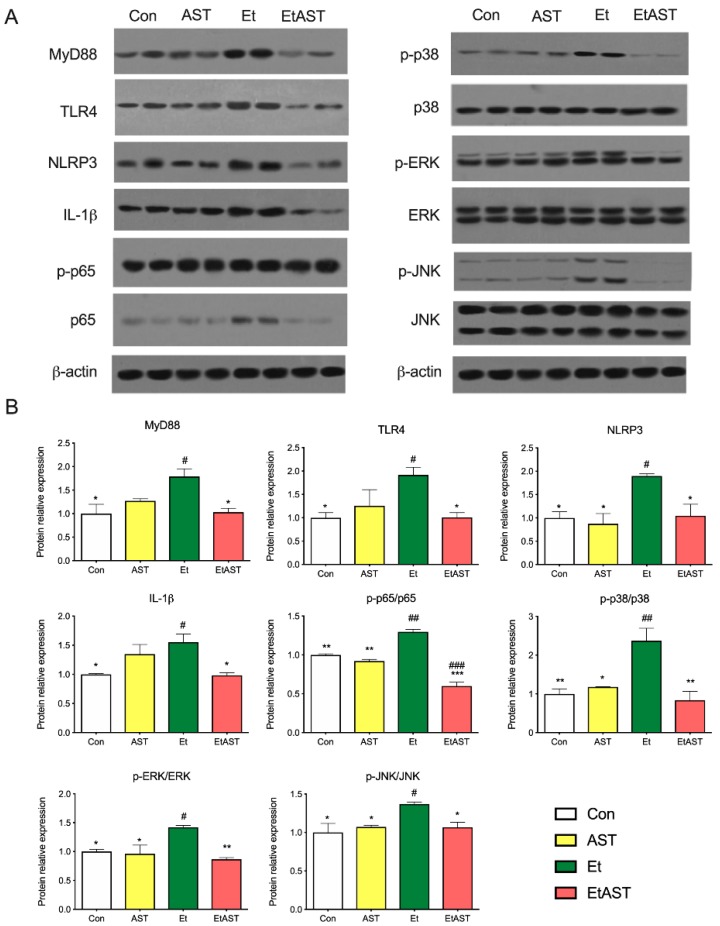
Hepatic protein expression levels of selected genes involved in the NOD-like pathway, Toll-like pathway, and chemokine pathway. The protein expression (**A**) and relative protein levels (**B**) were measured by western blot analysis. The relative protein levels were measured by Western blot analysis. Data was represented as means ± SD (*n* = 6). * *p* < 0.05 versus Et; ** *p* < 0.01 versus Et; # *p* < 0.05 versus Con; ## *p* < 0.01 versus Con; and ### *p* < 0.001 versus Con.

**Table 1 marinedrugs-17-00181-t001:** Summary of RNA-sequencing data.

Sample	Con	AST	Et	EtAST
Total reads (× 10^5^)	514.95 ± 6.89	546.02 ± 15.93	576.06 ± 21.01	690.85 ± 54.14
Total mapped reads (× 10^5^)	473.87 ± 6.11	501.50 ± 14.26	527.61 ± 19.32	632.12 ± 49.06
Mapped to reference genome %	92.02	91.85	91.59	91.5
Mapped to gene %	97.47	97.25	97.04	97.64
Mapped to exon %	95.35	95.67	95.56	96.24
Mapped to intergene %	2.55	2.76	2.72	2.37

Con: mice fed with fed with a Lieber–DeCarli liquid diet, Astaxanthin (AST): mice fed with a Lieber–DeCarli liquid diet and astaxanthin, Et: mice fed with an ethanol-containing Lieber–DeCarli liquid diet, EtAST: mice fed with ethanol-containing Lieber–DeCarli liquid diet and astaxanthin.

**Table 2 marinedrugs-17-00181-t002:** Functional annotation of transcriptome data in three public databases.

Database	Annotated	Percent
GO	21,430	97.06
KEGG	13,582	61.52
eggNOG	16,465	74.58
Ensembl	22,078	100

**Table 3 marinedrugs-17-00181-t003:** Statistics on the Kyoto Encyclopedia of Genes and Genomes (KEGG) pathway enrichment of DEGs between Con and Et.

Pathway ID	Pathway	Con up	Et up	P Value	FDR
ko04062	Chemokine signaling pathway	1	46	9.13 × 10^−11^	7.03 × 10^−9^
ko04612	Antigen processing and presentation	0	26	5.62 × 10^−9^	2.49 × 10^−7^
ko04621	NOD-like receptor signaling pathway	0	19	1.71 × 10^−7^	4.38 × 10^−6^
ko04650	Natural killer cell mediated cytotoxicity	0	32	3.38 × 10^−7^	7.80 × 10^−6^
ko04672	Intestinal immune network for IgA production	0	15	1.47 × 10^−6^	2.83 × 10^−5^
ko04610	Complement and coagulation cascades	0	23	2.04 × 10^−6^	3.62 × 10^−5^
ko04611	Platelet activation	0	27	6.64 × 10^−6^	1.10 × 10^−4^
ko04620	Toll-like receptor signaling pathway	1	21	6.48 × 10^−5^	7.87 × 10^−4^
ko04623	Cytosolic DNA-sensing pathway	0	16	7.96 × 10^−5^	9.20 × 10^−4^
ko04666	Fc gamma R-mediated phagocytosis	0	19	9.31 × 10^−5^	9.90 × 10^−4^

Con up: the DEGs which were up-regulated in control group, Et up: the DEGs which were up-regulated in ethanol group, FDR: false discovery rate.

**Table 4 marinedrugs-17-00181-t004:** Statistics on the KEGG pathway enrichment of DEGs between EtAST and Et.

Pathway ID	Pathway	Et up	EtAST up	P Value	FDR
ko04650	Natural killer cell mediated cytotoxicity	40	1	1.81 × 10^−12^	1.33 × 10^−10^
ko04621	NOD-like receptor signaling pathway	22	0	7.34 × 10^−10^	2.70 × 10^−8^
ko04062	Chemokine signaling pathway	43	1	3.38 × 10^−9^	9.34 × 10^−8^
ko04612	Antigen processing and presentation	26	0	5.62 × 10^−9^	1.38 × 10^−7^
ko04620	Toll-like receptor signaling pathway	28	1	8.00 × 10^−9^	1.77 × 10^−7^
ko04610	Complement and coagulation cascades	24	0	5.28 × 10^−7^	9.71 × 10^−6^
ko04672	Intestinal immune network for IgA production	15	0	1.47 × 10^−6^	2.3 × 10^−5^
ko04666	Fc gamma R-mediated phagocytosis	22	0	2.07 × 10^−6^	2.92 × 10^−5^
ko04611	Platelet activation	26	2	2.11 × 10^−6^	2.92 × 10^−5^
ko04640	Hematopoietic cell lineage	19	2	1.18 × 10^−5^	1.45 × 10^−4^

Et up: the DEGs which were up-regulated in ethanol group, EtAST up: the DEGs which were up-regulated in ethanol plus astaxanthin group, FDR: false discovery rate.

## Data Availability

The datasets generated and/or analyzed during the current study are available at NCBI project PRJNA524945 with accession number (SRR8689617-SRR8689631).

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
