# Peer review of "Comparative Transcriptome Analyses Provide Potential Insights into the Molecular Mechanisms of Astaxanthin in the Protection against Alcoholic Liver Disease in Mice"

_marinedrugs, 2019, doi:10.3390/md17030181_

Round 1

Reviewer 1 Report

In this study, Liu H and colleagues reported that effect of astaxanthin on the protection against alcoholic liver diseases in mice using comparative transcriptome analysis.

The experiment itself is carried out with careful treatments, but I think the approach is quite common, therefore not so much valuable information will be included in this manuscript. In general, the manuscript lacks novelty and  it seems that it does not extend our understanding.

Therefore, in the present state, this manuscript is inappropriate for publication.

1)    For Abstract and line 67, what dose “Con, AST, Et, and EtAST groups” stand for?

2)    For Table 3 and Table 4, a more detailed description is required.

3)    For References, journal names are missing in several places, for instance, from #14 to #18, and from #28 to #41.

4)    For Author Contributions, because there are authors of the same initials, I think that it is better to distinguish them.

5)    Line 57, what dose “AXT” stand for?

6)    Line 225, what dose “C” stand for?

7)    Line 286-287, please include this document, (Anders and Huber, 2010), in References.

8)    For Figure 4, what do “p-p65/p65”, “p-p38/p38”, “p-ERK/ERK” and “p-JNK/JNK” stand for?

9)    For animal experiments, how was the state of the animals during the animal experiment? For example, weight, feeding amount, blood biochemistry test, side-effect, and so on? What is the absorption rate of astaxanthin into the living body?

10) For Discussion or Conclusion, what is the new story when compared with previous studies?

11) Minor revisions are listed below.

1.       Line 56: “in vivo” should be italic.

2.       Line 84: Space is required between Table 2 and line 84.

3.       Line 105: “Figure 1b” should be “Figure 1B.”

4.       Lines 101-102: “(a) Venn diagram of gene counts expressed in the Con, AST, Et, and EtAST groups. (b) Number” should be “(A) Venn diagram of gene counts expressed in the Con, AST, Et, and EtAST groups. (B) Number.”

5.       Lines 133-134 : “(a) the NOD-like pathway, (b) Toll-like pathway, and (c) chemokines” should be “(A) the NOD-like pathway, (B) Toll-like pathway, and (C) chemokines.”

6.       Line 138: “Figure 3a” should be “Figure 3A.”

7.       Line 141: “astaxanthin” should be “AST.”

8.       Line 143: “Figure 3b” should be “Figure 3B.”

9.       Line 146: “astaxanthin” should be “AST.”

10.    Line 148: “Figure 3c” should be “Figure 3C.”

11.    Line 150: “astaxanthin” should be “AST.”

12.    Line 155: “astaxanthin” should be “AST.”

13.    Line 173: “astaxanthin” should be “AST.”

14.    Line 181: “in vivo” should be italic.

15.    Line 189: “Toll-like receptor” should be “TLR.”

16.    Line 196: “Toll-like receptors” should be “TLRs.”

17.    Line 204: “Figure 3a” should be “Figure 3A.”

18.    Line 209: “Figure 3 and 4” should be “Figures 3 and 4.”

19.    Line 214: “pathogen- PAMPs and DAMPs” should be “pathogen-PAMPs and -DAMPs.”

20.    Line 222: “interleukin-18 (IL-18)” should be “IL-18.”

21.    Line 315: IL-1b should be IL-1b.

22.    Figure 3: IL-1a, IL-1b, and NRLP1a should be IL-1a, IL-1b, and NRLP1, respectively.

23.    Figure 4: IL-1b should be IL-1b.

24.    Figure S2: TNF-a, IL-1a, IL-1b, and NRLP1a should be TNF-a, IL-1a, IL-1b, respectively.

Author Response

Thank you for your letter and for the reviewers’ comments concerning our manuscript entitledComparative transcriptome analyses provide potential insights into the molecular mechanisms of astaxanthin in the protection against alcoholic liver disease in mice” (marinedrugs-455602). Those comments are all valuable and very helpful for revising and improving our paper, as well as the important guiding significance to our researches. We have studied comments carefully and have made correction which we hope meet with approval. The main corrections in the paper and the responds to the reviewer’s comments are as flowing:

Point 1: For Abstract and line 67, what dose “Con, AST, Et, and EtAST groups” stand for?

Response 1:   Thank you very much for your comments. Considering your suggestion, the explanation of these aberrations (Con, AST, Et, and EtAST) were added in the abstract. Please see the abstract in the revised version.

Point 2: For Table 3 and Table 4, a more detailed description is required.

Response 2:  Thank you very much for your comments. According to your suggestion, we provide more detailed description in this part, we also add FDR value in the Table 3 and Table 4, please see line162 to 163 in the revised version.

Point 3: For References, journal names are missing in several places, for instance, from #14 to #18, and from #28 to #41.

Response 3: Thank you very much for your comments. We are so sorry for the mistakes. In the revised version, we add the journal names. Please see line 504 to 537 and line 558 to 586 in the revised version.

Point 4: For Author Contributions, because there are authors of the same initials, I think that it is better to distinguish them.

Response 4: Thank you very much for your comments. We used different initials for Huilin Liu (HL. L) and Huimin Liu (HM. L) in the revised version, please see line 467 to 469.

Point 5: Line 57, what dose “AXT” stand for?

Response 5: We are so sorry for the mistake. AXT should be AST, the aberration of astaxanthin. We corrected it in the revised version, pleases see line 75.

Point 6: Line 225, what dose “C” stand for?

Response 6:  We are so sorry for the mistake. It’s unwanted, we deleted it in the revised version.

Point 7: Line 286-287, please include this document, (Anders and Huber, 2010), in References.

Response 7: Thank you very much for your comment. We added the paper in references, please see line 585 in the revised version.

Point 8: For Figure 4, what do “p-p65/p65”, “p-p38/p38”, “p-ERK/ERK” and “pJNK/JNK” stand for?

Response 8: All these represented for the phosphorylation levels of p65, p38, ERK 1/2, JNK. It’s the ratio of phosphorylated protein to total protein.

Point 9:  For animal experiments, how was the state of the animals during the animal experiment? For example, weight, feeding amount, blood biochemistry test, side-effect, and so on? What is the absorption rate of astaxanthin into the living body?

Response 9: Thank you very much for your comment. The detailed results of animal experiment were reported in our previous paper, including the weight changes, plasma properties, and pathological morphology. Please see reference [12]. For the absorption rate of astaxanthin, we didn’t examine in the previous study. However, we will do this experiment in the future study. According to some previous publications, the efficiency of AST absorption from olive oil averaged 20% with individual samples having a range of 14 to 28% in male Holtzman albino rats 1, and the astaxanthin bioavailability in human plasma was confirmed with single dosage of 100 mg 2. Astaxanthin is a fat-soluble compound, with increased absorption when consumed with dietary oils. Taken together, its absorption is dependent on the accompanying dietary components.

1.     Clark R M, Yao L, She L, et al. A comparison of lycopene and astaxanthin absorption from corn oil and olive oil emulsions[J]. Lipids, 2000, 35(7):803-806.

2.     Chimsung N, Tantikitti C, Milley J E , et al. Effects of various dietary factors on astaxanthin absorption in Atlantic salmon [J]. Aquaculture Research, 2014, 45(10):1611-1620.

Point 10: For Discussion or Conclusion, what is the new story when compared with previous studies?

Response 10: Thank you very much for your comment. According to your comments, we revised the discussion and conclusion. In brief, compared with previous study, ours is the first report describing the comparative transcriptome analyses of AST in the protection against ALD. We also found that AST may prevent the progress of ALD through the chemokine signaling pathway, NOD-like receptor signaling pathway, and TLR signaling pathway. Please see line 449 to 452 and line 247 to 250.

Point 11: Minor revisions

1. Line 56: “in vivo” should be italic.

2. Line 84: Space is required between Table 2 and line 84.

3. Line 105: “Figure 1b” should be “Figure 1B.”

4. Lines 101-102: “(a) Venn diagram of gene counts expressed in the Con, AST, Et, and EtAST groups. (b) Number” should be “(A) Venn diagram of gene counts expressed in the Con, AST, Et, and EtAST groups. (B) Number.”

5. Lines 133-134 : “(a) the NOD-like pathway, (b) Toll-like pathway, and (c) chemokines” should be “(A) the NOD-like pathway, (B) Toll-like pathway, and (C) chemokines.”

6. Line 138: “Figure 3a” should be “Figure 3A.”

7. Line 141: “astaxanthin” should be “AST.”

8. Line 143: “Figure 3b” should be “Figure 3B.”

9. Line 146: “astaxanthin” should be “AST.”

10. Line 148: “Figure 3c” should be “Figure 3C.”

11. Line 150: “astaxanthin” should be “AST.”

12. Line 155: “astaxanthin” should be “AST.”

13. Line 173: “astaxanthin” should be “AST.”

14. Line 181: “in vivo” should be italic.

15. Line 189: “Toll-like receptor” should be “TLR.”

16. Line 196: “Toll-like receptors” should be “TLRs.”

17. Line 204: “Figure 3a” should be “Figure 3A.”

18. Line 209: “Figure 3 and 4” should be “Figures 3 and 4.”

19. Line 214: “pathogen- PAMPs and DAMPs” should be “pathogenPAMPs and -DAMPs.”

20. Line 222: “interleukin-18 (IL-18)” should be “IL-18.”

21. Line 315: IL-1b should be IL-1b. 22. Figure 3: IL-1a, IL-1b, and NRLP1a should be IL-1a, IL-1b, and NRLP1, respectively.

22. Figure 3: IL-1a, IL-1b, and NRLP1a should be IL-1a, IL-1b, and NRLP1, respectively.

23. Figure 4: IL-1b should be IL-1b.

24. Figure 2: TNF-a, IL-1a, IL-1b, and NRLP1a should be TNF-a, IL1a, IL-1b, respectively.

Response 11:  We are so sorry for so many mistakes in the manuscript. According to your comments, we correct all the mistakes in the revised version.

Reviewer 2 Report

Liu et al have submitted a nice manuscript on Alcoholic liver disease (ALD) is a major cause of chronic liver disease worldwide. They describe how it is a multi-complex process including a broad spectrum of hepatic lesions, from fibrosis to cirrhosis.  The focus of this paper is Astaxanthin (AST) a xanthophyll carotenoid, obtained from marine organisms and algae. It has the potential to alleviate hepatic inflammation and lipid dysmetabolism induced by ethanol. The authors performed comparative hepatic transcriptome analysis among the groups treated with or without AST via RNAseq. Gene enrichment analysis was conducted to identify impacted pathways. DE genes were verified by quantitative RT-PCR (qRT-PCR) and Western blot. The analyses revealed that the chemokine signaling pathway, antigen processing, and presentation pathway, nucleotide-binding and oligomerization domain (NOD)-like receptor signaling pathway, and Toll-like receptor signaling pathway enriched the most DEGs. This provide insights for the development of nutrition-related therapeutics for ALD.

Overall this is a nice manuscript that will be of interest to both scientists that work on marine derived compounds and those that work on ALD. The paper provides important input on the application of nutrition-related therapeutics for ALD.

I would recommend the following revisions before publication

1)      Details on the RNAseq needs to be better described – was PE or SE sequencing done.

2)      Which Illumina Hiseq instrument was used.

3)      In the interests of rigor and transparency in research, the RNAseq data should be submitted to a public data e.g. GEO.

4)      I am concerned that the data analysis for RNAseq does not include FDR analysis – which is the accepted standard today – a p-value of 0.05 is really not stringent enough for RNAseq – why did the authors not use FDR.

5)      Similarly for the data in Table 3 – why was a corrected p-value not provided ?

6)      The Q-PCR analysis is not adequately described. Details on the Q-PCR need to be improved. With the aim of providing transparent research and promoting rigor and transparency - further details need to be provided for the Q-PCR assays - QPCR assays - so that the data is MIQE compliant. Were the designed assays exon spanning ? - How can the authors be sure the amplicons derive from RNA and not genomic DNA ? - no DNAse treatment of RNA is mentioned ?. The authors do not state if melting curves were performed or not - this data could be included in supplemental materials. Authors should note the The MIQE Guidelines: Minimum Information for Publication of Quantitative Real-Time PCR Experiments http://clinchem.aaccjnls.org/content/55/4/611, and make sure that the data is compliant.

Author Response

Thank you for your letter and for the reviewers’ comments concerning our manuscript entitledComparative transcriptome analyses provide potential insights into the molecular mechanisms of astaxanthin in the protection against alcoholic liver disease in mice” (marinedrugs-455602). Those comments are all valuable and very helpful for revising and improving our paper, as well as the important guiding significance to our researches. We have studied comments carefully and have made correction which we hope meet with approval. The main corrections in the paper and the responds to the reviewer’s comments are as flowing:

Point 1: Details on the RNAseq needs to be better described – was PE or SE sequencing done.

Response 1:   Thank you very much for your comments. PE sequencing was done in our study. We add this detail in the revised version, please see line 382.

Point 2: Which Illumina Hiseq instrument was used.

Response 2:  Thank you very much for your comments. Illumia Hiseq Xten was used for RNA-seq. We add this detail in the revised version, please see line 382 .

Point 3: In the interests of rigor and transparency in research, the RNAseq data should be submitted to a public data e.g. GEO.

Response 3: Thank you very much for your comments.  The datasets generated and/or analyzed during the current study are available at NCBI project. The raw data is about 77Gb, so it is still in uploading now, we will provide SRP number as soon as possible.

Point 4: I am concerned that the data analysis for RNAseq does not include FDR analysis – which is the accepted standard today – a p-value of 0.05 is really not stringent enough for RNAseq – why did the authors not use FDR.

Response 4: Thank you very much for your comments.  Considering your suggestion, we used FDR to correct p-value in DEG analyse. Please see line 162 to 163 .

Point 5: Similarly for the data in Table 3 – why was a corrected p-value not provided ?

Response 5: Thank you very much for your comments.  According to your suggestion, we provide FDR values in Table 3 and Table 4.

Point 6: The Q-PCR analysis is not adequately described. Details on the Q-PCR need to be improved. With the aim of providing transparent research and promoting rigor and transparency - further details need to be provided for the Q-PCR assays - QPCR assays - so that the data is MIQE compliant. Were the designed assays exon spanning ? - How can the authors be sure the amplicons derive from RNA and not genomic DNA ? - no DNAse treatment of RNA is mentioned ?. The authors do not state if melting curves were performed or not - this data could be included in supplemental materials. Authors should note the The MIQE Guidelines: Minimum Information for Publication of Quantitative Real-Time PCR Experiments http://clinchem.aaccjnls.org/content/55/4/611, and make sure that the data is compliant.

Response 6:  Thank you very much for your comment. We performed Q-PCR assays according with MIQE compliant during experiments to ensure the data is compliant. Moreover, reverse-transcribed was used the PrimeScript RT Reagent Kit with gDNA eraser to avoid genomic DNA pollution, and the specific primers designed spanning exon to distinguish DNA and RNA, and inhibit DNA amplification. For the melting curves, we performed under the condition: 95 °C for 60 s followed by 55 °C for 30 s and then 95 °C for 30 s. The data of melting curves were provided in supplemental materials (Figure S4).

Reviewer 3 Report

This manuscript describes investigation of astaxanthin (AST) effects on alcoholic liver disease using hepatic transcriptome analysis among mouse groups treated with or without AST. The results are clear and support the conclusions.

Abstract should indicate this study is based on mice experiments although the title indicates so.

In abstract, abbreviations (Con, AST, Et, EtAST, DEG) should be explained.

Author Response

Thank you for your letter and for the reviewers’ comments concerning our manuscript entitledComparative transcriptome analyses provide potential insights into the molecular mechanisms of astaxanthin in the protection against alcoholic liver disease in mice” (marinedrugs-455602). Those comments are all valuable and very helpful for revising and improving our paper, as well as the important guiding significance to our researches. We have studied comments carefully and have made correction which we hope meet with approval. The main corrections in the paper and the responds to the reviewer’s comments are as flowing:

Comments: Abstract should indicate this study is based on mice experiments although the title indicates so. In abstract, abbreviations (Con, AST, Et, EtAST, DEG) should be explained.

Response: Thank you very much for your comments. According to your suggestion, we revised the abstract. The description of mice experiment and the explanation of abbreviations (Con, AST, Et, EtAST, DEG) were added into the abstract. Please see the abstract in revised version.

Round 2

Reviewer 1 Report

I thank the authors’ reply to my comments.

The revised manuscript looks fine. However, I have a few comments before the acceptance of this study.

1)    To make it easier for the readers to see the intent of the experiments just by looking at the figures or tables, how about describing abbreviations and/or simple results in legends? For instance, in Table 1, I think that it is better to explain Con, AST, Et, and EtAST, and in Tables 3 & 4, it is better to explain Con up, Et up, EtAST up, and FDR.

2)    Minor revisions are listed below.

1.       Lines 87-88: Please space between Table 2 and line 88.

2.       Line 132: “pathway(ko04621)” should be “pathway (ko04621)”.

3.       Line 135: “pathway(ko04621), and the chemokine signaling pathway(ko04062)” should be “pathway (ko04621), and the chemokine signaling pathway (ko04062)”.

4.       Line 195: “in vitro” should be italic.

5.       Line 196: “in vivo” should be italic.

6.       Line 200: “liver injury[12]” should be “liver injury [12]”.

7.       Line 279: “astaxanthin” should be “AST”.

8.       Lines 315-316: For“an endogenous reference gene,” please align fonts.

9.       Line 355: For “SRP ( )”, the parentheses are blank.

10.    Line 398: “Journal of surgical research” should be “J Surg Res”.

11.    Line 399: “Journal of agricultural” should be “J Agric Food Chem”.

12.    Line 403: “The Journal of nutritional biochemistry” should be “J Nutr Biochem”.

13.    Line 406: “Hepatology research” should be “Hepatol Res”.

14.    Line 426: Journal is missing. World J Hepatol?

15.    Line 428: “Annual Review of Pathological Mechanical Disease” should be “Ann Rev of Pathol”.

16.    Line 430: “International journal of physiology, pathophysiology& pharmacology” should be “Int J Physiol Pathophysiol Pharmacol”.

17.    Lines 432-433: “Relevance of the NLRP3 inflammasome in the Pathogenesis of Chronic Liver Disease. Frontiers in immunology” should be “Relevance of the NLRP3 inflammasome in the pathogenesis of chronic liver disease. Front Immunol”.

18.    Line 435: “Mediators of Inflammation” should be “Mediators Inflamm”.

19.    Line 440: “Journal of interferon cytokine research” should be “J Interferon Cytokine Res”.

20.    Line 448: “World journal of gastroenterology” should be “World J Gastroenterol”.

Author Response

Thank you for your letter and for the reviewers’comments concerning our manuscript entitledComparative transcriptome analyses provide potential insights into the molecular mechanisms of astaxanthin in the protection against alcoholic liver disease in mice” (marinedrugs-455602). Those comments are all valuable and very helpful for revising and improving our paper, as well as the important guiding significance to our researches. We have studied comments carefully and have made correction which we hope meet with approval. The main corrections in the paper and the responds to the reviewer’s comments are as flowing:

Point 1:To make it easier for the readers to see the intent of the experiments just by looking at the figures or tables, how about describing abbreviations and/or simple results in legends? For instance, in Table 1, I think that it is better to explain Con, AST, Et, and EtAST, and in Tables 3 & 4, it is better to explain Con up, Et up, EtAST up, and FDR. 

Response 1: Thank you so much for your comments. Considering your suggestion, we add the explanation of the abbreviations in Table 1, Table 3 and Table 4. 

Point 2:Minor revisions

1. Lines 87-88: Please space between Table 2 and line 88.

2. Line 132: “pathway(ko04621)” should be “pathway (ko04621)” 

3. pathway(ko04062)” should be “pathway (ko04621), and the chemokine signaling pathway (ko04062)”. 

4. Line 195: “in vitro” should be italic. 

5.Line 196: “in vivo” should be italic. 

6. Line 200: “liver injury[12]” should be “liver injury [12]” 

7. Line 279: “astaxanthin” should be “AST”. 

8.Lines 315-316: For“an endogenous reference gene,” please align fonts. 

9. Line 355: For “SRP ( )”, the parentheses are blank.

10. Line 355: For “SRP ( )”, the parentheses are blank.Line 398: “Journal of surgical research” should be “J Surg Res”. 

11. Line 399: “Journal of agricultural” should be “J Agric Food Chem”. 

12. Line 403: “The Journal of nutritional biochemistry” should be “J Nutr 

Biochem”. 

13. Line 406: “Hepatology research” should be “Hepatol Res”. 

14. Line 426: Journal is missing. World J Hepatol

15. Line 428: “Annual Review of Pathological Mechanical Disease” 

should be “Ann Rev of Pathol”. 

16. Line 430: “International journal of physiology, pathophysiology& pharmacology” should be “Int J Physiol Pathophysiol Pharmacol”. 

17. Lines 432-433: “Relevance of the NLRP3 inflammasome in the Pathogenesis of Chronic Liver Disease. Frontiers in immunology” should be “Relevance of the NLRP3 inflammasome in the pathogenesis of chronic liver disease. Front Immunol”.

18. Line 435: “Mediators of Inflammation” should be “Mediators Inflamm”. 

19. Line 440: “Journal of interferon cytokine research” should be “J Interferon Cytokine Res”.

20. Line 448: “World journal of gastroenterology” should be “World J Gastroenterol”. 

Response11: We are so sorry for so many mistakes in the manuscript. According to your comments, we correct all the mistakes in the revised version.

Reviewer 2 Report

I commend the authors on responding to comments raised during peer review - the article is now acceptable for publication. I would strongly encourage providing the GEO accession for the sequencing data prior to publication.

Author Response

Thank you for your letter and for the reviewers’comments concerning our manuscript entitledComparative transcriptome analyses provide potential insights into the molecular mechanisms of astaxanthin in the protection against alcoholic liver disease in mice” (marinedrugs-455602). Those comments are all valuable and very helpful for revising and improving our paper, as well as the important guiding significance to our researches. We have studied comments carefully and have made correction which we hope meet with approval. The main corrections in the paper and the responds to the reviewer’s comments are as flowing:

Comment:I commend the authors on responding to comments raised during peer review - the article is now acceptable for publication. I would strongly encourage providing the GEO accession for the sequencing data prior to publication.

Response: Thank you so much for your comment. The datasets generated and/or analyzed during the current study are available at NCBI project PRJNA524945 with accession number (SRR8689617-SRR8689631), and all the raw data will be released on 31 Mar, 2020. Moreover, the Musmusculusgenome were used as reference genome in current study. So we used the primers described in previous studies for the validation study.